# Immunophenotyping of a Stromal Vascular Fraction from Microfragmented Lipoaspirate Used in Osteoarthritis Cartilage Treatment and Its Lipoaspirate Counterpart

**DOI:** 10.3390/genes10060474

**Published:** 2019-06-21

**Authors:** Denis Polancec, Lucija Zenic, Damir Hudetz, Igor Boric, Zeljko Jelec, Eduard Rod, Trpimir Vrdoljak, Andrea Skelin, Mihovil Plecko, Mirjana Turkalj, Boro Nogalo, Dragan Primorac

**Affiliations:** 1Srebrnjak Children’s Hospital, HR-10000 Zagreb, Croatia; lzenic@bolnica-srebrnjak.hr (L.Z.); turkalj@bolnica-srebrnjak.hr (M.T.); nogalo@bolnica-srebrnjak.hr (B.N.); 2Specialty Hospital St. Catherine, HR-49120 Zabok, Croatia; damir.hudetz@svkatarina.hr (D.H.); igor.boric@svkatarina.hr (I.B.); zeljko.jelec@svkatarina.hr (Z.J.); Eduard.rod@svkatarina.hr (E.R.); trpimir.vrdoljak@svkatarina.hr (T.V.); andrea.skelin@svkatarina.hr (A.S.); mihovil.plecko@gmail.com (M.P.); 3Clinical Hospital Sveti Duh, HR-10000 Zagreb, Croatia; 4School of Medicine, Josip Juraj Strossmayer University of Osijek, HR-31000 Osijek, Croatia; 5School of Medicine, University of Rijeka, HR-51000 Rijeka, Croatia; 6School of Medicine, University of Split, HR-21000 Split, Croatia; 7Genos Glycoscience Research Laboratory, HR-10000 Zagreb, Croatia; 8Catholic University of Croatia, HR-10000 Zagreb, Croatia; 9University of New Haven, West Haven, CT 06516, USA; 10Department of Biotechnology, University of Rijeka, HR-51000 Rijeka, Croatia; 11Eberly College of Science, The Pennsylvania State University, University Park, PA 16803, USA

**Keywords:** lipoaspirate, microfragmented lipoaspirate, immunophenotyping, endothelial progenitors, pericytes, supra-adventitial adipose stromal cells

## Abstract

Osteoarthritis (OA) is a degenerative joint disease accompanied by pain and loss of function. Adipose tissue harbors mesenchymal stem/stromal cells (MSC), or medicinal signaling cells as suggested by Caplan (Caplan, 2017), used in autologous transplantation in many clinical settings. The aim of the study was to characterize a stromal vascular fraction from microfragmented lipoaspirate (SVF-MLA) applied for cartilage treatment in OA and compare it to that of autologous lipoaspirate (SVF-LA). Samples were first stained using a DuraClone SC prototype tube for the surface detection of CD31, CD34, CD45, CD73, CD90, CD105, CD146 and LIVE/DEAD Yellow Fixable Stain for dead cell detection, followed by DRAQ7 cell nuclear dye staining, and analyzed by flow cytometry. In SVF-LA and SVF-MLA samples, the following population phenotypes were identified within the CD45^−^ fraction: CD31^+^CD34^+^CD73^±^CD90^±^CD105^±^CD146^±^ endothelial progenitors (EP), CD31^+^CD34^−^CD73^±^CD90^±^CD105^−^CD146^±^ mature endothelial cells, CD31^−^CD34^−^CD73^±^CD90^+^CD105^−^CD146^+^ pericytes, CD31^−^CD34^+^CD73^±^CD90^+^CD105^−^CD146^+^ transitional pericytes, and CD31^−^CD34^+^CD73^high^CD90^+^CD105^−^CD146^−^ supra-adventitial-adipose stromal cells (SA-ASC). The immunophenotyping profile of SVF-MLA was dominated by a reduction of leukocytes and SA-ASC, and an increase in EP, evidencing a marked enrichment of this cell population in the course of adipose tissue microfragmentation. The role of EP in pericyte-primed MSC-mediated tissue healing, as well as the observed hormonal implication, is yet to be investigated.

## 1. Introduction

Ever since the discovery that human adipose tissue is a rich source of mesenchymal stem/stromal cells (MSC) and is bestowed with therapeutic potential after culture expansion and differentiation [1,2,3], autologous adipose tissue has been explored for its regenerative properties in many clinical settings. MSC are contained within the stromal vascular fraction (SVF)—the aqueous portion obtained after enzymatic digestion of lipoaspirate (LA)—together with other cells of the differentiated or multipotent features. The success of tissue regeneration has been attributed to the activated MSC excreting bioactive molecules that inhibit scar formation and stimulate angiogenesis. The immunomodulatory and regenerative properties of MSC acting in a paracrine manner provoked the re-naming of these cells as medicinal signaling cells (MSC) [4,5].

Lipogems^®^ technology has emerged as a platform that yields an intact SVF and MSC of a high therapeutic potential (reviewed in [6]), obtained by the microfragmentation of autologous fat tissue [7]. The use of such microfragmented lipoaspirate (MLA) has reached a fruitful application in osteoarthritis (OA), one of the leading musculoskeletal disorders in the adult population worldwide [8]. A prospective non-randomized interventional study using a single intra-articular injection of autologous MLA resulted in a successful osteoarthritic cartilage treatment, as evidenced by an increased glycosaminoglycan content and an improved mechanical axis of the patient lower extremities [9]. The microfragmentation of adipose tissue is considered to empower tissue regeneration by boosting the secretory ability of MSC. As shown recently, the secretome of microfragmented adipose tissue is abundantly rich in cytokines and angiogenic factors [10]. Although clinical implementation of MLA has expanded and yielded impressive results in tissue healing (especially in orthopedia and reconstitutive surgery), a more precise cellular characterization to correlate explanatory aspects is still called for. As demonstrated previously by flow cytometry analysis of human lipoaspirate, the SVF is comprised of endothelial progenitor (EP) cells (CD31^+^CD34^+^CD146^+^), endothelial mature (EM) cells (CD31^+^CD34^−^CD146^±^), pericytes (CD31^−^CD34^−^CD146^+^), supra-adventitial-adipose stromal cells (SA-ASC) (CD31^−^CD34^+^CD146^−^) and the transitional pericytes (TP) (CD31^−^CD34^+^CD146^+^), exhibiting various expressions of the CD73, CD90 and CD105 mesenchymal markers [11]. A recent paper by Vezzani et al. analyzed adipose-derived perivascular MSC obtained by microfragmentation, and documented a marked domination of pericytes over the SA-ASC, as compared to autologous lipoaspirate-derived SVF [10]. In this study, we applied a more comprehensive approach by polychromatic flow cytometry in order to unambiguously define the multipotent cellular composition and ratios in the SVF from LA or MLA, based on the previous papers [10,11,12].

## 2. Materials and Methods

### 2.1. Patients

The study involved twelve patients with OA (six males and six females, aged 30–81) receiving intra-articular knee injection of autologous MLA in the Specialty Hospital St. Catherine (Zabok, Croatia), as described in Hudetz et al. [8]. The LA and MLA patient samples were delivered to the Srebrnjak Children’s Hospital (Zagreb, Croatia) and stored overnight at room temperature (RT) protected from light before further processing [13,14]. The study was approved by the local Institutional Review Board (No. EP 001/2016) and the hospital Research Ethics Committee (No. 11/2017).

### 2.2. Stromal Vascular Fraction Isolation

To maximize the SVF yield from both sample sources and properly prepare cells for flow cytometry, we equally treated LA and MLA samples with 1% collagenase type I in Dulbecco’s Modified Eagle Medium (D-MEM) (both from Sigma-Aldrich, Saint Louis, MO, USA) in a shaking bath at 37 °C for 45 min. After 1:2 dilution with 2% fetal bovine serum (Biosera, Nuaille, France) in D-MEM (Sigma-Aldrich), samples were filtrated through a 100 µm-cell strainer (BD Falcon, Corning, NY, USA) and centrifuged at 300 g for 10 min at RT. Supernatants were discarded and cell pellet resuspended in 1 mL of the VersaLyse solution (Beckman Coulter, Miami, FL, USA). After 10 min, samples were filtered through a 40 µm-cell strainer (BD Falcon, Corning, NY, USA), centrifuged at 300 g for 10 min at RT and the cell pellet resuspended in D-MEM (Sigma-Aldrich). The cells were counted on the Sysmex XT1800 counter (Sysmex, Kobe, Japan).

### 2.3. Flow Cytometry

SVF cells isolated from LA (SVF-LA) or MLA (SVF-MLA) were stained using a Duraclone SC dry reagent prototype tube (kindly provided by Beckman Coulter, Miami, FL, USA). The Duraclone SC tube is a polychromatic reagent that allows the identification of MSC subpopulations based on the use of antibodies specific for the cell surface markers: CD31, CD34, CD45, CD73, CD90, CD105, and CD146, labeled with PB, ECD, APC-AF750, PE, FITC, CD45-PC7, and PC5.5 fluorochromes, respectively, and Live/Dead Yellow Fixable Stain (ThermoFisher, Waltham, MA, USA) for 20 min at RT protected from light, fixed with 2% paraformaldehyde (Electron Microscopy Sciences, Hatfield, PA, USA) in phosphate-buffered saline (PBS; Sigma-Aldrich), washed, permeabilized with PermWash (BD Biosciences, San Jose, CA, USA) and the cell nuclei stained with the DRAQ7 dye (BioStatus, Shepshed, UK). Forward scatter (FCS) data files were analyzed using the FlowLogic software (Inivai Technologies, Mentone, Australia). The data about reproducibility of staining with Duraclone SC tube can be found in Appendix A. More details about the instrument configuration, daily quality control, reagents used and data analysis can be found in Appendix A.

### 2.4. Statistical Analysis

For data that passed the normality test, a parametric paired *t*-test, unpaired *t*-test, or one-way analysis of variance (ANOVA) with Sidak’s multiple comparison test was used. For data that failed the normality test, a nonparametric Wilcoxon test was applied to calculate the statistics (GraphPad Prism 6 for Windows; GraphPad Software, Inc., San Diego, CA, USA). A *p*-value < 0.05 was considered statistically significant.

## 3. Results

### 3.1. Immunophenotyping Analysis of Stromal Vascular Fraction from Lipoasirate and Microfragmented Lipoasirate Samples by Polychromatic Flow Cytometry

The gating strategy and subpopulation determination are shown in Figure 1A–D, Figure 2A–D, Figure 3A–D and Appendix A. The inclusion of a DNA dye was crucial to identify and select nucleated cells from the mixture of the remaining cell debris, red blood cells and oil residues. CD45^−^ and CD45^+^ SVF cells were determined in both lipoaspirate (SVF-LA) and its microfragmented lipoaspirate counterpart (SVF-MLA), and viability determined. The viability of nucleated CD45^−^ was 94% ± 1.54% for LA and 94.8% ± 1.2% for MLA, and the viability of nucleated CD45^+^ cells was 93.7% ± 1.2% for LA and 94.4% ± 1.6% for MLA (Figure 4A,B). Only live cells were selected for further analysis (Figure 1D).

We identified the five CD45^−^ subpopulations of SVF-LA and SVF-MLA as being: CD31^+^CD34^+^CD73^±^CD90^±^CD105^±^CD146^±^ EP, CD31^+^CD34^−^CD73^±^CD90^±^CD105^−^CD146^±^ EM, CD31^−^CD34^−^CD73^±^CD90^+^CD105^−^CD146^+^ pericytes, CD31^−^CD34^+^CD73^±^CD90^+^CD105^−^CD146^+^TP, and CD31^−^CD34^+^CD73^high^CD90^+^CD105^−^CD146^−^ supra-adventitial adipose stromal cells (Figure 2A–D and Figure 3A–D). In all the samples, the average percentage of EM and TP was always below 2% of the total nucleated cells, and therefore the data from these populations were not included in further analysis. The four major determined phenotypes are summarized in Table 1.

### 3.2. Stromal Vascular Fraction from Microfragmented Lipoasirate Significantly Differs in Cell Content from Stromal Vascular Fraction from Lipoasirate

The relative amounts of the four subpopulations were expressed as a percentage of total nucleated cells. The percentage of EP was significantly higher (Figure 5A) while the percentage of SA-ASC and leukocytes was significantly lower in SVF-MLA compared to SVF-LA (Figure 5B,D). The percentage of pericytes showed a donor-dependent variation and did not result in a significant difference between SVF-LA and SVF-MLA (Figure 5C).

Given the literature data on pericyte-mediated tissue regeneration via activated MSC [6,10], we were particularly interested in the ratio between pericytes and SA-ASC. As shown in Figure 6A, the pericytes/SA-ASC ratio was significantly higher in SVF-MLA versus SVF-LA samples. Interestingly, samples with the highest pericytes/SA-ASC ratio originated from male patients, and this was significantly higher compared to female patients (Figure 6D). The EP/pericytes and EP/SA-ASC ratios were also significantly higher in SVF-MLA compared to SVF-LA (Figure 6B,C), however, it did not show sex-related difference in the former (Figure 6E,F).

## 4. Discussion

In an attempt to contribute to the growing need to understand the adipose tissue MSC used in therapeutic intervention, we investigated the cell content of the MLA product applied for the treatment of OA patients and its lipoaspirate counterpart [9,15]. We described five CD45^−^ subpopulations of SVF-LA and SVF-MLA, here described as being: CD31^+^CD34^+^CD73^±^CD90^±^CD105^±^CD146^±^EP cells, CD31^+^CD34^−^CD73^±^CD90^±^CD105^−^CD146^±^ EM cells, CD31^−^CD34^−^CD73^±^CD90^+^CD105^−^CD146^+^ pericytes, CD31^−^CD34^+^CD73^±^CD90^+^CD105^−^CD146^+^ TP and CD31^−^CD34^+^CD73^high^CD90^+^CD105^−^CD146^−^SA-ASC. The relative proportions of the determined nucleated populations, with the EP percentage prevailing over SA-ASC and pericytes, as well as the expressions of the CD73 and CD105 mesenchymal markers differ from what has been reported previously for non-microfragmented adipose tissue [11,12,16]. Although some previous papers reported CD105 positivity on freshly isolated SVF-LA cells [16,17], we found only a weak CD105 expression on pericytes and SA-ASC of either SVF-LA or SVF-MLA samples. The observed differences could be ascribed to different adipose tissue extraction methods (abdominoplasty versus lipoaspiration) and also the analysis approach, which in our case involves the calculation of the subpopulation frequency of all nucleated events and not only of the non-leukocyte fraction.

With respect to microfragmented adipose tissue, we observed even more pronounced differences in the proportions of the determined populations than seen in the lipoaspirate counterparts. To our knowledge, this is the first comprehensive immunophenotyping of the SVF populations from MLA used for therapeutic intervention. The analysis of SVF cell composition performed so far by flow cytometry has been carried out by means of expressing a single marker-positivity, e.g., as CD34^+^ or CD146^+^ populations within nucleated CD45^−^ cells in MLA [7]. Given that these are not population-specific phenotypes, and that different subpopulations share certain markers, this approach does not provide a precise content of the SVF subpopulations and their ratios. In a recent paper, Vezzani et al. applied a similar approach to ours and reported a substantial pericyte increase after microfragmentation with respect to the SA-ASC share [10]. We also found a marked domination of pericytes over SA-ASC, yet, our pericyte/SA-ASC ratio was even more pronounced (five times higher median value). Of note, the authors did not specify the use of the DNA stain to exclude non-nucleated events from the analysis, which might have caused a higher background and could have contributed to double negative events influencing the observed proportions. Moreover, the authors focused only on pericytes and SA-ASC while leaving out the other cell populations from the analysis.

Pericytes and SA-ASC, the presumed in vivo progenitors of MSC [18,19], have gained the most attention in interpreting the medicinal effects of MSC used in clinical applications. However, after microfragmentation, we surprisingly found the most prominent enrichment of EP cells, which outnumbered both pericytes and SA-ASC. When MLA was applied as an autologous intra-articular treatment in patients with late-stage knee OA, it improved clinical and functional outcomes in 85% of the patients, who had before the study been candidates for a total knee replacement surgery [15]. The fact that the applied MLA EP outnumbered the other SVF cells imposes their inevitable involvement in the observed effects of MLA-mediated cartilage treatment. EP represent important players implicated in vascularization as a prerequisite for wound repair and tissue regeneration [20,21]. Several in vitro and in vivo studies have demonstrated proliferative, proangiogenic and vasculogenic effects of MSC–EP interactions [22,23,24,25], as well as differentiation commitment [26,27]. A crosstalk between MSC–EP, which can involve both paracrine mechanism and/or cell–cell contact (via platelet-derived growth factor and its receptor (PDGF–PDGFR), and Notch-signaling) [23,28], seems to occur in different conditions of tissue inflammation [29], suggesting that their interplay and beneficial effects might also be accomplished in cartilage tissue repair and regeneration. We, therefore, hypothesized that a plethora of cytokines, chemokines and growth factors, i.e., bioactive molecules hitherto ascribed to the activated MSC, acting as medicinal signaling cells [5], perhaps stem from a synergistic effect of the pericyte–MSC–EP interactions [15].

We furthermore found a marked decrease of the CD45^+^ leukocyte fraction in MLA, confirming the previous reports on the reduction of proinflammatory elements by microfragmentation technology [7]. A lower inflammatory cytokine source presumably contributes to the repair process at the site of tissue injury. However, the reduction of the CD45- population per se is not enough for the observed repair effects which have been evidently ascribed to the MSC-induced vasculogenic and other factors [7].

The notion that in our hands the EP-pericyte ratio, as well as the pericyte-SA-ASC ratio, differed between men and women implies that the presumed in vivo precursors of the MSC, as well as EP, might be under the influence of sex hormone(s). In line with this, we noticed that in the paper by Vezzani et al. [10], the sample source originated exclusively from female subjects and resembles our female data, which besides technical and analytical disparities might partially explain the above-mentioned lower pericyte/SA-ASC ratio in the cited paper. Interestingly, we did not observe sex-related difference between EP and pericytes, the two most prevalent populations in microfragmented products. The hormonal regulation of hematopoietic stem cells has already been evidenced [30], and further attempts should be focused on the hormone-specific pathways of their adipose MSC counterparts. It has been reported that estrogens may act upon the production of cytokines and growth factors in human MSC as well as the level of circulating EP (reviewed in [31]). The results of the in vivo studies are yet to explore the influence of autocrine estrogen production by MSC on their therapeutic applications [32].

## 5. Conclusions

Taking into account our results, we have continued our clinical study enrolling a higher number of participants, which will elucidate the observed sex disparity phenomenon. Ultimately, our goal is to associate this clinical aspect with the secretome of the applied MLA product and the success of the treatment outcome. Our results offer scientific support necessary to corroborate promising results of clinical applications of the microfragmented product by elucidating structural niches that hold functional impact. The pericyte–EP crosstalk has been documented intensely [33,34], and we believe that the role of this axis in tissue regeneration cannot be neglected [35]. Given the intriguing result we found, we emphasize the importance of further investigation of these interactions in pericyte-mediated MSC activation and, furthermore, in provoking angiogenic and anti-inflammatory features of MSC in cartilage repair [36,37]. We also found a marked decrease in the CD45^−^ leukocyte fraction, confirming the previous reports on the reduction of proinflammatory elements by microfragmentation technology [7].

## Figures and Tables

**Figure 1 genes-10-00474-f001:**
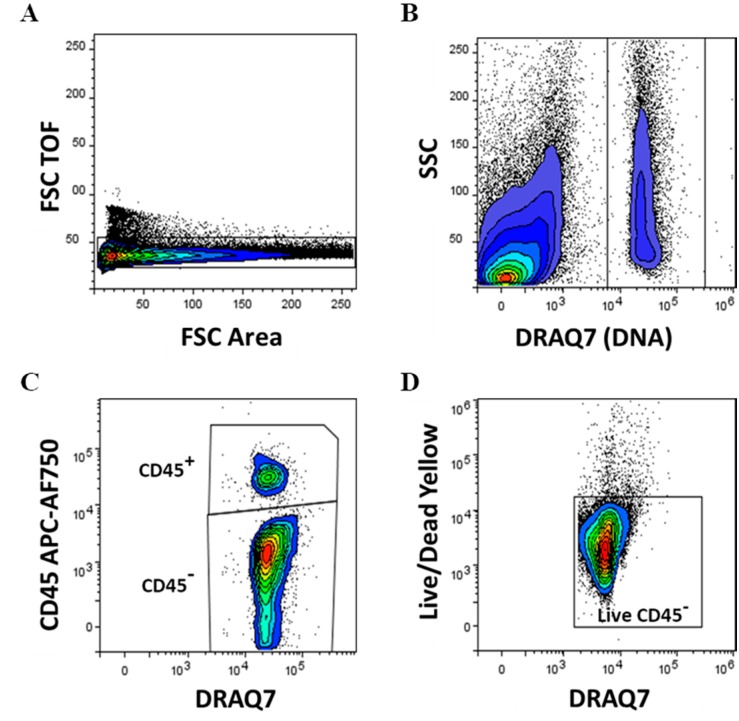
Gating strategy, expression of the CD45 cell surface marker and cell viability in stromal vascular fraction (SVF) cells obtained from lipoaspirate (SVF-LA) or its microfragmented lipoaspirate (SVF-MLA) counterpart. Singlet cell selection was performed on forward scatter (FCS) time of flight (TOF) and FSC area signals density plot (**A**). Only events from nucleated cells were analyzed further, based on the DNA-binding DRAQ7 dye-positivity and side scatter (SSC) (**B**). Nucleated CD45^+^ and CD45^−^ cells were discerned (**C**), and live CD45^−^ cells (**D**) and CD45^+^ cells determined.

**Figure 2 genes-10-00474-f002:**
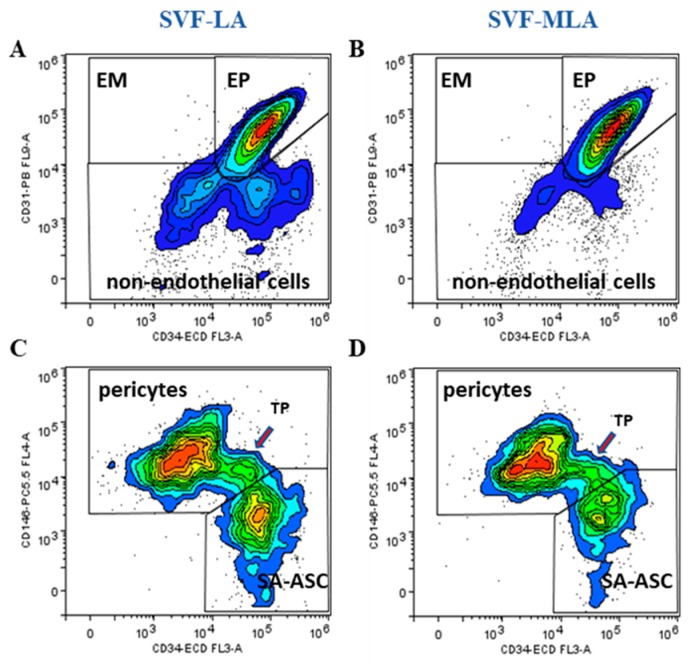
Nucleated CD45^−^ live cells were analyzed for the presence of the CD31 and CD34 lineage specific surface markers. Endothelial mature (EM), endothelial progenitor (EP) and non-endothelial cells were identified as CD31^+^CD34^−^, CD31^+^CD34^+^ and CD31^−^CD34^−^ cells, respectively (**A**,**B**). Non-endothelial cells were analyzed further for CD146 expression in combination with CD34 to discern pericytes, transitional pericytes (TP) and supra-adventitial-adipose stromal cells (SA-ASC) in SVF-LA and SVF-MLA samples (**C**,**D**).

**Figure 3 genes-10-00474-f003:**
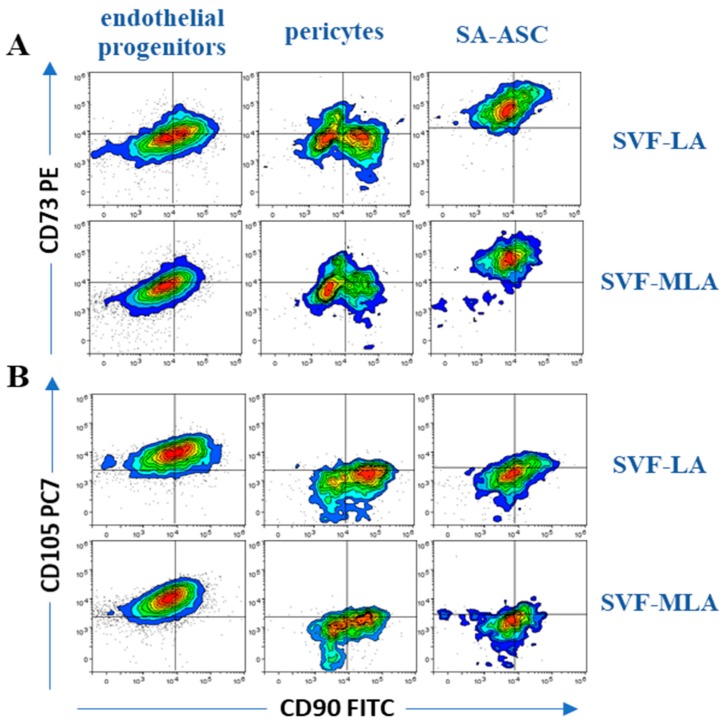
Expression of mesenchymal stem cell markers in SVF cells obtained from SVF-LA or its SVF-MLA counterpart. The co-expression of the mesenchymal-associated markers CD90 vs CD73 (**A**) and CD90 vs CD105 (**B**) are shown for the three progenitor populations: endothelial progenitors, pericytes and SA-ASC. One representative experiment is shown. Summarized results of the determined immunophenotypes for the three progenitor populations and leukocytes are shown in Table 1, *n* = 12.

**Figure 4 genes-10-00474-f004:**
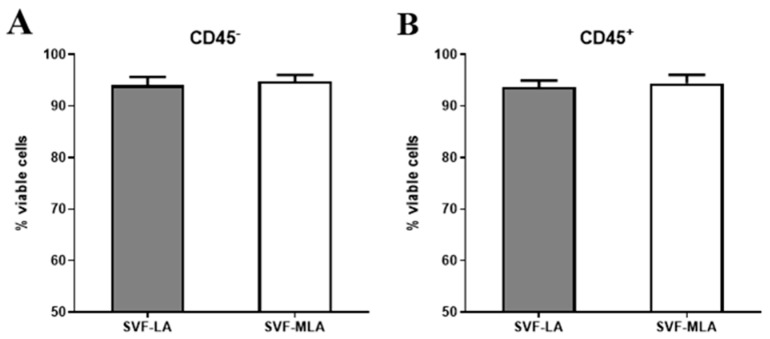
Cell viability in CD45^−^ and CD45^+^ SVF cells obtained from SVF-LA or its SVF-MLA counterpart.

**Figure 5 genes-10-00474-f005:**
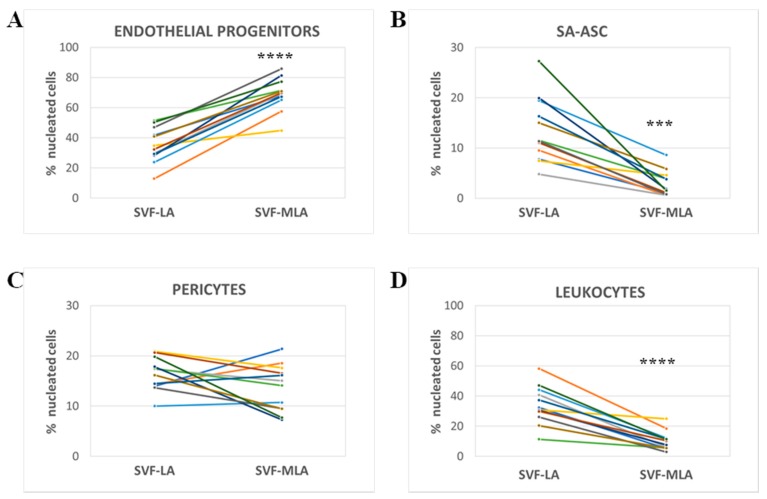
Comparative analysis of four cell subpopulations in SVF obtained from SVF-LA or its SVF-MLA counterpart. The quantitative data of the polychromatic flow cytometry analysis show the difference in the percentage of the three progenitor cell types (**A**–**C**) and leukocytes (**D**) between the SVF-LA and SVF-MLA of each donor. *p*-values from *p* < 0.01; (***) *p* < 0.001; (****) *p* < 0.0001; *n* = 12.

**Figure 6 genes-10-00474-f006:**
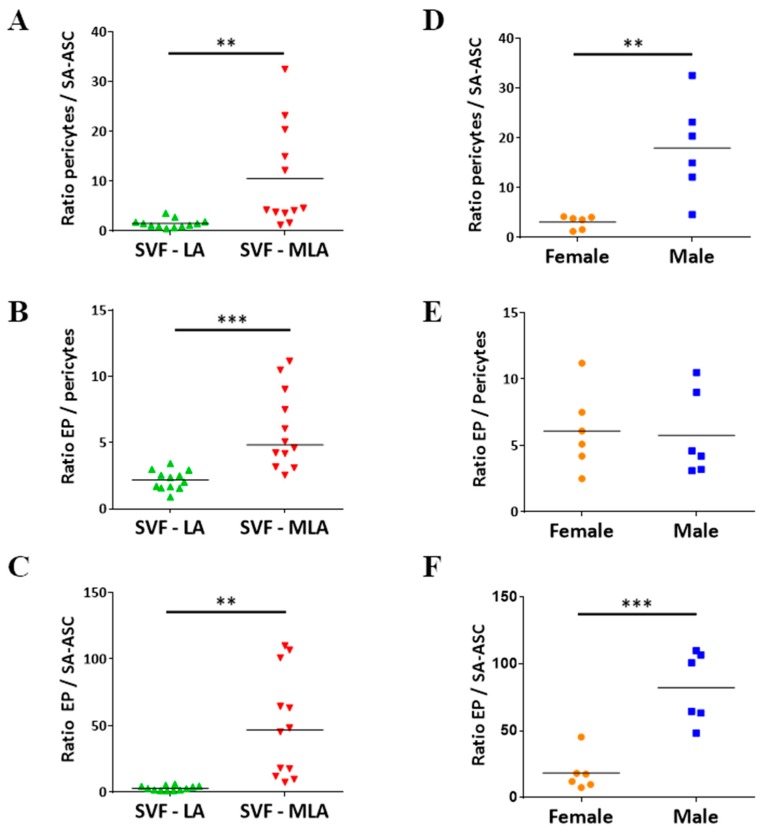
Comparative analysis of four cell subpopulations in SVF cells obtained from (SVF-LA or its SVF-MLA counterpart. Graphs **A**–**C** show a difference in the ratios of the three progenitor subpopulations between SVF-LA and SVF-MLA. Differences in the ratios of the three progenitor subpopulations in the SVF-MLA samples between female and male patients are shown in graphs **D**–**F**. *p*-values: (**) *p* < 0.01; (***) *p* < 0.001; *n* = 12.

**Table 1 genes-10-00474-t001:** Summarized results of the determined immunophenotypes and their frequency within nucleated SVF cells for the three progenitor populations and leukocytes.

Immunophenotype	Lineage Markers	Mesenchymal Stem/Stromal Cell (MSC) Markers	Frequency within Nucleated SVF Cells	
			LA	MLA
Endothelial progenitors	CD45^−^CD31^+^CD34^+^CD146^±^	CD73^±^CD90^±^CD105^±^	13–51%	45–89%
Pericytes	CD45^−^CD31^−^CD34^−^CD146^+^	CD73^±^CD90^+^CD105^−^	10–21%	7–21%
SA-ASC	CD45^−^CD31^−^CD34^+^CD146^−^	CD73^high^CD90^±^CD105^−^	5–27%	1–9%
Leukocytes	CD45^+^CD31^−^CD34-CD146^−^	CD73^−^CD90^−^CD105^−^	11–58%	3–25%

± stands for a variable expression (negative and positive cells overlap).

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
