# Peer review of "Immunophenotyping of a Stromal Vascular Fraction from Microfragmented Lipoaspirate Used in Osteoarthritis Cartilage Treatment and Its Lipoaspirate Counterpart"

_genes, 2019, doi:10.3390/genes10060474_

Round 1
Reviewer 1 Report
This paper is an interesting study of immunophenotyping stromal fraction from micro-frgmented lipoaspirate. It utilises flow cytometry to look at a range of markers.
I think this study is of interest to a number of scientists. However, i would like to recommend a few improvements to the flow cytometry data.
- MIFlowCyt guidelines should be adhered to, as this will give more information about the flow cytometry allowing readers to intrepret the data easier and potentially reproduce it.
- There should be more information given about the cytometer. The cytometer should be listed in the methods and not merely named in a figure. More information about the cytometer could be added to supplementary information. This information should include the filters on the Navios.
- I would like to see more information about the gating on Figure 3 and 4. How were the gates set? This is particularly important for intrepretation of figure 4.
Author Response
Pllease find attached document

Reviewer 2 Report
In the manuscript submitted by Polancec et al., they outline their findings of phenotypic comparisons of LA vs MLA SVF fraction using a prototype flow panel. Although their efforts to better quantitate SVF fractions is highly commendable, and much needed in the field, there are several concerns that need to be addressed.
Major:
1)Figure 3 shows the gating strategy from one patient. Can the authors comment on the reproducibility of the assay from one patient to another? How about multiple samples from the same patient?
2)There should be some explanation in the methods about how the gates were set, particularly for Figure 4. It is concerning when you have a quadrant in which the center falls right in the middle of a population (see Figure 4A lower left graph). Also, stating a phenotype like CD73±CD90±CD105± is really confusing. Does ± mean mixed populations, dim expression, inconsistent expression, or other? Using positive controls (like cultured MSCs) and negative controls will help the reader interpret the flow data better.
3)The Discussion and Conclusion sections are inadequate.
a) Most of the 1st part of the discussion is a summary of the abstract and introduction.
b) Since this was a phenotypic characterization of SVF fractions isolated via different methods and no functional or outcome data was presented, the authors really need to be more careful making assumptions on what is the “active component” of the cell product. Until the different cell types/ratios can be tested in trials, statements regarding therapeutic benefit should be minimized. Since the aim of the study was to comparison phenotypic differences between LA and MLA, the audience would benefit greatly if the authors could discuss their results to what has been shown previously with references 9,10, and 11.
c) The data shown in Fig 5D is actually really interesting where the % of leukocytes is significantly decreased in the MLA samples. In the conclusions, the authors state that the enrichment of particular building blocks might contribute to a more successful OA treatment. An alternative explanation is that the removal of leukocytes makes the entire product more “efficacious”. A discussion of the implications of leukocyte removal would be helpful.
d) The conclusion should state their take home message and how these results advance the field. How should researches move forward with these results?
Minor:
1)On line 59, 60: “…yields an intact SVF and MSC of a high therapeutic potential…”. The term high therapeutic potential is really vague so the authors should either add a reference to support the statement to be more specific or take out.
2)According to Reference #6, the Lipogems technology appears not to need enzymatic digestion, but in the methods on line 81, it states that MLA samples were digested with 1% collagenase. Can the authors please clarify?
3)Line 121, 122: the authors state that the viability for both CD45+ and CD45- cells isolated under the two conditions was 95%. Is that really the case? In Figure 2A, the means look different between the LA and MLA samples.
4)Table 2 is not really a “results” table but a list of phenotypes. Would be very helpful for the audience to see means and ranges of all the phenotypes collected from the 12 patients with OA.
5)Line 230: I believe the authors mean CD45+ leukocytes instead of CD45- as written?
Round 2
Reviewer 2 Report
The authors have satisfied most of my concerns. I would add that Table 1 is much improved but it still would be helpful to add CD45 as a lineage marker, whereas the leukocytes would be CD45+ and the other cells CD45-.
Author Response
Dear reviewer,
thank you for your kind suggestion. We agree adding of CD45 to lineage markers column is making table better and makes table more imformative.
Best wishes and kind regards,
Denis Polancec